# Regulation of Physiological Barrier Function by the Commensal Microbiota

**DOI:** 10.3390/life13020396

**Published:** 2023-01-31

**Authors:** Simon McArthur

**Affiliations:** Institute of Dentistry, Faculty of Medicine & Dentistry, Queen Mary University of London, Blizard Institute, 4, Newark Street, London E1 2AT, UK; s.mcarthur@qmul.ac.uk; Tel.: +44-20-7882-7133

**Keywords:** gut epithelium, epidermis, blood-brain barrier, microbiota, metabolites

## Abstract

A fundamental characteristic of living organisms is their ability to separate the internal and external environments, a function achieved in large part through the different physiological barrier systems and their component junctional molecules. Barrier integrity is subject to multiple influences, but one that has received comparatively little attention to date is the role of the commensal microbiota. These microbes, which represent approximately 50% of the cells in the human body, are increasingly recognized as powerful physiological modulators in other systems, but their role in regulating barrier function is only beginning to be addressed. Through comparison of the impact commensal microbes have on cell–cell junctions in three exemplar physiological barriers—the gut epithelium, the epidermis and the blood–brain barrier—this review will emphasize the important contribution microbes and microbe-derived mediators play in governing barrier function. By extension, this will highlight the critical homeostatic role of commensal microbes, as well as identifying the puzzles and opportunities arising from our steadily increasing knowledge of this aspect of physiology.

## 1. Introduction

We live in a microbial world; microbes evolved long before eukaryotes, let alone humans, and are likely to exist long after our species has become extinct. As such, human physiology is highly adapted and responsive to microbial signals, whether these are deleterious or have been co-opted into homeostatic roles. The composition of the microbiota is highly plastic and responsive to numerous intrinsic and extrinsic factors, including, but not limited to, nutrition, aging, immune function and circadian rhythm [1,2]. As such the influence of microbes upon the body can also vary substantially across physiological and pathological states. Deciphering the nature of these changes and their underlying mechanisms is thus of great importance in building our understanding of normal physiology and disease, of how microbial actors can affect these processes beyond their classical roles in infection, and ultimately for therapeutic exploitation.

The importance of host–microbe communication is particularly apparent at the numerous biological barriers within the body, whether those separating the external environment from the internal, such as the epidermis or the gut, lung, bladder and uterine epithelia, or those dividing the circulation and local tissue environments, e.g., the blood–brain, blood–ocular, blood–thymus or blood–testis barriers. These systems are fundamental drivers of physiological homeostasis, enabling the formation of separate tissue microenvironments and governing communication between them. Whilst different barrier systems have their own characteristics, a key underlying feature held in common between them is the presence of molecular cell–cell junction complexes, the structures that ultimately govern barrier integrity. This review will examine the impact of microbial actors upon cell–cell junctions, focussing on three key example barriers, the gut epithelium, the skin epidermis and the blood–brain barrier, barriers that are associated with a high or low microbial load, or, under physiological conditions at least, sterility.

## 2. The Gut Microbiota and the Intestinal Epithelium

While we have known of the existence of the gut microbiota for many years, the more recent identification that it can exert major modifying influences upon host systems has had a profound impact on our understanding of physiology, forcing a re-evaluation of organisms away from the picture of an individual towards that of a holobiont, an intricate and far from fully understood ecosystem of host and commensal microbes. Probably the most significant example of this is the human gut microbiome, a highly complex ecosystem consisting of approximately 10^14^ microbial cells, roughly equal to the number of human cells in the entire body [3] and comprising around one hundred times more genes than are found in the human genome itself [4]. Given this complexity, it is perhaps understandable that our grasp of the influences gut microbes can exert upon the body is in its infancy. It is nonetheless clear that gut microbes potently regulate the integrity and function of the gastro-intestinal tract epithelial barrier.

### 2.1. The Structural Barriers of the Gut

The epithelial cells lining the gastrointestinal tract form one of the largest surfaces in the body, with an estimated area of around 30 m^2^ [5], and perform a critical role governing nutrient uptake and protecting the internal environment from potentially harmful luminal contents. These ostensibly contradictory functions are achieved through an intricate set of junctional and transporter molecules expressed within the epithelial cells themselves, aided and supported by a complex network of interacting structural and immune systems, which together provide a dynamic physical and immunological barrier.

The intestinal epithelial barrier itself comprises several interacting elements. The outermost layer is a secreted coat of mucus that serves to protect the epithelium from potential mechanical, chemical and biological insults deriving from luminal contents such as food antigens, bacterial components and toxins, and environmental pollutants [6]. This mucus is a complex mix of water, proteins (predominantly the gel-like mucins, notably mucin 2 [7]), lipids and electrolytes secreted by the epithelial goblet cells [8], exocrine cells present throughout the gastro-intestinal tract but particularly abundant within the colonic mucosa [9]. The gel-like properties of mucus are key to its function as they enable both lubrication of food as it moves along the gastro-intestinal tract, avoiding mechanical injury, and the trapping of microbes and debris and their consequent removal by peristalsis-stimulated mucus flow.

In addition to these physical properties, the intestinal mucus is enhanced by the presence of numerous immune actors, including a range of potent anti-microbial peptides, e.g., defensins, LL-37, and lysozyme [10], immunoglobulin A [11] and co-opted bacteriophage viruses [12], that together provide a powerful anti-bacterial immune defence. Although mucus is present throughout the gastro-intestinal tract, it varies in composition and structure through different compartments. For example, the ileal mucus layer is discontinuous and uni-lamellar, presumably to enable digestive enzyme secretion and facilitate nutrient uptake [13], while the colonic mucus is divided into an anchored juxta-epithelial layer that is relatively inaccessible to bacteria and a more fluid luminal layer that is widely colonised by commensal bacteria, themselves a barrier to further colonisation by potential pathogens [14,15].

Beneath the mucus layer, columnar epithelial cells form the first cellular layer of the mucosa and serve as the major restrictive element controlling communication between the gastro-intestinal tract lumen and the internal environment. Intestinal epithelial cells are notable for their expression of robust and highly organised intercellular junctional components, consisting of multiple interacting proteins (see Table 1), together forming the mutually reinforcing tight junctions (TJs), junctional adhesion molecules (JAMs), adherens junctions (AJs) and desmosomes, sometimes collectively termed the apical junctional complex (AJC). Functionally, this structure serves as a tight permeability barrier, essentially limiting passive paracellular diffusion of luminal contents across the intestinal wall to water and electrolytes only (Figure 1).

A substantial proportion of the intestinal epithelial barrier function of the AJC is provided by the TJ complexes, found at the apex of the lateral membranes of the cells [16]. These multi-protein structures centre upon interacting tetra-spanning transmembrane proteins, including occludin, tricellulin and one or more of the 27 members of the claudin family [17,18], which then interact with intracellular scaffold proteins such as zonula occludens (ZO)-1, -2 or -3 [19], themselves further associating with the actin cytoskeleton [20]. Both claudins and occludin form homotypic (claudins and occludin) and heterotypic (claudin subtypes) interactions with their counterparts on neighbouring cells through their extracellular domains [18,21], bringing epithelial cells into close apposition and effectively sealing the intestinal barrier.

Whilst expression of ZO-1, JAMs, occludin (found primarily at bicellular junctions [22]) and tricellulin (found at junctions of three or more cells [23]) are found at more or less constant levels throughout the gastro-intestinal tract epithelium, claudin family member expression differs markedly between different regions of the tract and indeed along the crypt-to-apex axis of individual villi (reviewed in [24]). Not all claudins form a strict barrier, with certain family members generating paracellular channels permeable to cations (claudins-2, -10b, -15, -16 and -21) and anions (claudins-10a and -17), whilst claudin-2 forms a paracellular water channel [25]. Moreover, individual claudin members can differ in their function depending on whether they form homotypic or heterotypic cell–cell associations, e.g., claudin-4 forms a tight homotypic link, but serves as a chloride channel when co-expressed alongside claudin-8 [26]. These claudin subtype differences are thought to directly underlie the varying barrier properties of different regions of the gastro-intestinal tract epithelium; electrophysiological resistance is greatest in the duodenum, a region associated with the ‘tight’ claudins -1, -3, -5 and -8, in contrast to the jejunum and ileum, in which the relatively ‘leaky’ claudins-2, -7 and -12 predominate [27].

Tight junctions are not static structures, and are subjected to a range of processes governing their formation and disruption [28,29], with the actions of JAM family members in particular being well characterised examples. JAM proteins, single transmembrane domain members of the wider immunoglobulin superfamily, are enriched at the site of TJs in the intestinal epithelium, where they are important regulators of TJ physiology [30]. While the prototypical JAM, JAM-A, has a broad subcellular distribution, it is highly enriched at the site of TJs where it is thought to enable TJ assembly. Blocking of JAM-A functionality, either by immunoneutralisation [31] or inhibition of expression [32], reduced recruitment of TJ components to the AJC, impairing epithelial cell barrier function without necessarily inhibiting membrane apposition. Intricate molecular studies have shown that JAM-A plays a major role in the maturation of TJ complexes, acting via recruitment and facilitation of protein kinase C ι activity to promote an environment favouring TJ component recruitment and stabilisation, as reviewed in [33].

While important, TJs are not the sole governors of intestinal epithelial barrier integrity. Found more basally in the lateral membrane of the epithelial cells [34], AJs are a second source of structural support for the intestinal epithelium. AJs are composed of an alternative complex of transmembrane proteins, the Ca^2+^-dependent cadherins and their intracellular binding partners the catenins [35]. Cadherins, in the intestinal epithelium primarily E-cadherin [36], are single transmembrane spanning proteins with multiple extracellular Ca^2+^-binding subdomains that form homophilic interactions with cadherins on the surface membrane of neighbouring cells [37]. E-cadherin then binds through its intracellular domain to a complex consisting of p120 catenin, β-catenin and α-catenin, which itself binds the actin cytoskeleton [35]. Both TJs and AJs are linked via filamentous actin projections to the peri-junctional actomyosin ring, a belt-like structure composed of actin and myosin II protein that encompass the apical pole of the epithelial cell [38]. Despite providing only a weak direct contribution to barrier resistance, the relatively dynamic and short-lived AJs are critical in establishing the cell–cell contacts needed for TJ formation and maturation; indeed, mouse models of AJ depletion are characterised by extensive intestinal epithelial disruption, haemorrhage and embryonic lethality [39].

The third major structural component to the intestinal epithelial barrier is the presence of desmosomes, sometimes termed maculae adherentes, that are found most basally within the AJC. These are multi-protein complexes that provide support and aid resistance to mechanical stress, particularly important in the gastro-intestinal tract given the substantial forces the tissue is subjected to by both intestinal contents and peristaltic contractions [40]. Cell–cell interactions through desmosomes are mediated by members of two characteristic transmembrane cadherin families, the desmogleins and desmocollins, which draw areas of neighbouring cell lateral membranes together and serve as a platform for the assembly of proteinaceous cytoplasmic dense plaques [41]. The cytoplasmic tails of desmoglein and desmocollin cadherins associate with plakoglobin and plakophilin family members to form the outer dense plaque (named after its appearance in electron micrographs) [42]. This in turn binds the N-terminal head of desmoplakin which couples to intermediate filaments of the cytoskeleton, forming the inner dense plaque [43] and stabilising inter-epithelial junctional integrity.

### 2.2. Microbial Influences on the Intestinal Epithelial Barrier Structures

As mentioned above, the gut, and particularly the intestines, are host to a vast and diverse microbiota, shown to potently modulate epithelial barrier function directly through effects upon cell–cell junctions. This is most clearly demonstrated by analyses of gnotobiotic or germ-free mice, animals born and raised in a sterile environment and thus completely lacking a microbiota. That such animals have profoundly altered gastro-intestinal tracts has long been clear [44], but more recent molecular analysis of the germ-free intestine is beginning to reveal details of the gut microbes’ mediatory role(s) in shaping intestinal epithelial barrier function.

Whilst not strictly mediated through cell–cell junctions, it is notable that even the intestinal mucus layer is modified by actions of the gut microbiota. Mice born and raised under germ-free conditions show subtle changes in the colonic mucus; lamination of the mucus is still present, but the inner layer is significantly more permeable than that of conventional animals [45], a change reversed by reconstitution of the gut microbiota. This microbial influence on the colonic mucus structure may be relevant to pathology as both Crohn’s disease [46] and colorectal cancer [47] have long been associated with disruption in mucus lamination. The contribution of microbial influences to these pathologies is an area of active investigation, but, notably, consumption of probiotic bacteria has been shown to increase intestinal mucus secretion [48].

Beyond the mucus layer, however, there is strong evidence that the junctional proteins strengthening the epithelia themselves are subject to microbial influence. Studies of germ-free mice have revealed changes in the expression patterns of several major junctional component proteins, although there is considerable inconsistency between studies, suggesting that the effects of host genotype and/or environment still play an important role in the intestinal epithelial development, and cautioning against over-emphasis of the gut microbiota. For example, germ-free mice have been reported as exhibiting structural changes in the AJC, with around a third of epithelial cells having broader and shorter AJs and a lack of desmosomes [49]. In contrast, other studies have found no differences in the TJs or desmosomes between gnotobiotic and conventionally raised animals [50], while still others have shown the intestinal epithelium to be less permeable in germ-free mice [51,52], with increased expression of the TJ components claudin-1 and occludin, and greater paracellular transfer of luminal tracers to the bloodstream [51]. Although these differences between studies are confusing, they may represent the effects of microbe removal upon epithelial developmental processes as well as more immediate consequences.

As an alternative to germ-free development, studies have investigated the consequences of treatment with antibiotics as a tool to remove intestinal microbes. Here again, however, data are mixed. Some studies have shown removal of gut microbes to increase gut permeability, e.g., administration of the Gram negative-targeting antibiotic polymixin E to adult mice with disrupted TJ and desmosomal architecture, alongside reduced claudin-1, occludin and ZO-1 expression and a resultant bacteraemia [50], or oral administration of the more broad-spectrum vancomycin resulting in increased gut permeability, reduced claudin-4 expression and signs of inflammatory change [53]. In contrast, removal of murine intestinal bacteria with a neomycin/bacitracin mix significantly reduced gut permeability and up-regulated expression of ZO-1, JAM-A and occludin in the ileum and ZO-1 and claudins-3 and -4 in the colon [54]. It may be that these conflicting findings are due to the varying microbial specificities of the different antibiotics used, and that the overall effect of any given treatment regime will depend upon which bacterial species were present originally and the biological activities of organisms resistant to the chosen antibiotics.

In contrast to the discord in the literature regarding the impact of germ-free development, reports of the effect of gut microbe re-colonisation upon the intestinal epithelium of germ-free mice are much more consistent. Numerous studies have reported the normalisation of junctional proteins or of overall epithelial tightness following colonisation with specific strains of intestinal [49,51,55,56,57] or oral microbes [58], emphasising the importance of the microbiota in maintaining normal intestinal epithelial function, even if the precise mechanism(s) underlying these effects may still be unclear.

A number of studies have investigated the actions of probiotics, live bacteria that are consumed by an individual with the aim of re-colonising their gastro-intestinal tract, upon intestinal epithelial permeability, both in response to exogenous lesioning (reviewed in detail elsewhere [59]) or in the native state. Given the difficulties in interrogating mechanisms in vivo, the majority of studies have made use of immortalised epithelial cells, primarily the Caco-2 colorectal adenocarcinoma epithelial line, to decipher these effects. For example, treatment of these cells with *Lactobacillus* sp. bacterial cultures increased transepithelial electrical resistance and expression of numerous TJ and desmosomal components, including occludin, ZO-1, ZO-2 and plakoglobin, although, notably, none of the studied claudin family members were affected [56]. Importantly, similar results have been seen in vivo in experiments where healthy volunteers were intraduodenally fed with *Lactobacillus plantarum*, an intervention that rapidly resulted in enhanced expression of both TJ-associated occludin and ZO-1 in the duodenal enterocytes [60]. Analogous experiments using Caco2 cells, strongly have suggested that these effects are dependent upon the pattern recognition receptor toll-like receptor 2, TLR2 [55,60], highlighting the role of innate immune signalling pathways in governing the intestinal epithelial barrier.

Many microbial components, such as lipopolysaccharides, peptidoglycans or lipoteichoic acids, are potent activators of members of the toll-like receptor (TLR) family of pattern recognition receptors [61], with a number of family members having been shown to influence intestinal epithelial permeability. For example, stimulation of TLR2 leads to activation of protein kinase C isoforms α and δ, which in turn cause ZO-1 redistribution and an increase in barrier electrical resistance in vitro [55,62] and in vivo [60]. Similarly, soluble material from a number of probiotic strains has been shown to regulate TJ expression in Caco2 cells via TLR2-, TLR6- or TLR10-mediated activation of protein kinase C [63]. This link between epithelial permeability and pattern recognition receptors may not be wholly unexpected—one of the most fundamental functions of the intestinal epithelium is to protect the internal environment from invasion and colonisation by potentially pathogenic bacteria, hence the existence of the complex and potent mucosal immune system [64]. Given the wide range of microbe-derived structural molecules found in the intestinal lumen, development of a sensitive system for their detection, one that can be linked to the functional enhancement of the epithelial barrier, would seem an appropriate and likely evolutionary adaptation.

The ability to disrupt epithelial barrier function is a feature of several pathogenic strains of bacteria, driven by the production of protein enterotoxins. For example, *Clostridium perfringens* enterotoxin binds to both claudins-3 and -4, preventing their incorporation into TJs and thereby weakening epithelial integrity both in vitro [65] and in vivo [66]. Similarly, enteropathogenic *Escherichia coli*, but notably not non-pathogenic *E. coli* strains, produce a secreted effector protein EspF that phosphorylates occludin and stimulates its translocation from TJs into the cytoplasm, markedly enhancing intestinal permeability [67,68]. Such toxic actions have since been extended to other potentially pathogenic bacteria, including *Clostridium difficile*, *Helicobacter pylori*, *Campylobacter jejuni*, *Campylobacter concisus*, and *Salmonella typhimurium* (reviewed in detail in [69]), highlighting the potential of pathogenic bacteria as drivers, or possibly even initiators, of intestinal disease through their actions upon cell–cell junctional proteins.

An alternative pathway through which host epithelial cells respond to microbial elements, again particularly noted in conditions of disease, is via the protease activated receptors, PARs. These proteins, of which four family members (PAR1-4) are known, are seven transmembrane domain, G protein-coupled receptors bearing a tethered N-terminal activating sequence that can be exposed upon protease cleavage, leading to receptor activation and multiple downstream signalling events [70]. While all four PAR members have been identified at different locations in the gastro-intestinal tract, the majority of studies have focused on the roles of PAR1 and PAR2 and their involvement in intestinal pathologies [71]. Nonetheless, there is some evidence that gut microbes may influence PAR activity in physiological conditions, leading to changes in gut wall structure; germ-free mice exhibit altered vasculature of the villi, an effect that appears to be driven by the interaction of microbial proteases with PAR1 [72]. Less is known about whether and how microbe-induced PAR activity could modify cell–cell junctions, although in vitro analyses have shown PAR2 activation, by synthetic ligands [73] or by exogenous administration of mast cell tryptase [74], to disrupt and suppress expression of occludin, claudin-1 and ZO-1 but enhance that of claudin-2 by activation of autophagic processes [73]. Given that diverse gut microbes are known to secrete proteases capable of activating PARs, it seems highly possible that such molecules play a role in physiological maintenance of intestinal epithelial cell–cell junctions.

Sensitivity of the intestinal epithelium to microbe-derived molecules extends beyond the recognition of structural microbial components and toxins in pathological situations however, with the products of microbial metabolism also serving as physiological regulators of intestinal epithelial permeability. Some of the earliest studies to investigate the link between bacterial metabolites and the intestinal epithelial barrier focussed on the role of indole metabolites, products of microbial tryptophan degradation [75]. In particular, the indole derivative indole-3-propionic acid was shown to bind to and activate the pregnane X receptor, acting to promote barrier function indirectly through regulation of murine Tlr4 expression [75]. Notably, pregnane X receptor null mice displayed significantly reduced levels of occludin, ZO-1 and E-cadherin mRNA, effects that could be ameliorated by deletion of the Tlr4 gene [75], again highlighting both the complexity of the systems governing intestinal epithelial barrier function, and the importance of indirect actions of gut microbial metabolites upon immune defences.

Following these initial discoveries, a wider range of microbial metabolites have been investigated as modifiers of intestinal epithelial tight junction structure. For example, in vitro screening analysis of metabolites known to differ between healthy and dysbiotic mice identified both TJ stabilising (taurine, tryptamine and L-homoserine) and TJ disrupting (acetyl-proline, spermine, putrescine) molecules, with the effects of taurine and putrescine being validated in vivo [76]. Interestingly, the identified stabilising molecules were effective in countering TJ disruption caused by several different stimuli, suggestive of common downstream executive pathways for TJ disruption that might be being targeted.

A large number of studies have focussed on the regulatory actions of short chain fatty acids (SCFAs) upon the intestinal epithelium. These molecules, primarily acetate, propionate and butyrate, are produced to millimolar levels in the gut lumen by microbial fermentation of insoluble dietary carbohydrates and serve as a major energy source for intestinal epithelial cells [77,78]. Beyond this role however, there is considerable evidence that SCFAs can regulate cell–cell junction components, although the mechanisms involved are generally less clear-cut, perhaps unsurprisingly given that butyrate alone has been shown to act as a histone deacetylase inhibitor [79], to stabilise and thereby potentiate HIF-1α signalling [80] and to act directly through its own G protein-coupled receptors, FFAR2, FFAR3 and HCAR2 [81].

Multiple studies have indicated that SCFAs can alter epithelial permeability, with butyrate and propionate generally having been shown to improve barrier function both in vitro [82,83,84] and in vivo [80,85,86]. Results for acetate tend to be more mixed, with some studies reporting either neutral [82] or even detrimental [84] effects of this SCFA upon the intestinal epithelial barrier, perhaps reflecting the different complement of receptors activated by acetate in comparison with other SCFAs [81]. Mechanistic studies of the beneficial effects of SCFAs upon the intestinal barrier have identified the TJs as their primary downstream target, with few analyses made of potential effects upon AJs or desmosomes. Notably however, studies identifying TJs as targets have also highlighted the complexity and redundancy of the signalling networks governing their interactions with SCFAs. For example, butyrate has been shown to up-regulate claudin-1 expression via HIF-1α stabilisation [87,88], to upregulate and reorganise occludin and ZO-1 expression [89,90,91] via mobilisation of AMPK [92](incidentally, an effect not seen with acetate or propionate treatment [90]), to upregulate claudin-3, occludin and ZO-1 through HCAR2 [93] and to inhibit histone deacetylase activity, leading via STAT3 activation and increased production of the IL-10 receptor alpha subunit to suppression of the ‘leaky’ claudin protein, claudin-2 [80,86]. This profound diversity in mechanisms through which SCFAs can govern intestinal epithelial barrier function both alludes to the importance of these molecules in epithelial physiology, and emphasises the position of the gut microbiota, the primary source of SCFAs, in regulating the same.

Regulation of intestinal permeability is not only influenced by microbial metabolic waste products. For example, N-acyl homocysteine lactones (AHLs) are bacterial molecules best characterised for their role in quorum sensing—inter-bacterial communication that at its most basic allows for the monitoring of population density. Many different AHLs have been identified in the human gut [94], with unsaturated 3-oxo-C12:2 homoserine lactone being both the most conserved across species and the most abundant. Notably, this AHL has been shown to protect and stabilise intestinal epithelial TJs in vitro, preventing TNFα/IFNγ-induced ubiquitination and degradation of occludin and tricellulin [95]. Interestingly 3-oxo-C12:2 homoserine lactone is selectively lost in inflammatory bowel disease [94], potentially directly linking microbial actors with disease-associated epithelial barrier disruption.

While the complexity of interactions between the gut microbiota and the intestinal epithelium is clearly indicated from the studies described above, it is almost certain that our understanding of the list of microbial factors that can regulate barrier function is incomplete. Over 200 gut microbial metabolites/products can be detected in faeces [96,97], and the potential biological functions of only a fraction have been studied. Nonetheless, the gut microbiota is clearly a significant regulator of intestinal barrier function. This physiological importance may have relevance to pathology as well; both microbial dysbiosis and a disrupted intestinal epithelial barrier are features of several major diseases, including Crohn’s disease, ulcerative colitis and gastro-intestinal cancer. Investigation of the molecular factors regulating microbe–host communication in the gut under normal conditions, alongside how they may vary in disease, offers the distinct opportunity to identify new approaches for therapeutic intervention.

## 3. The Skin Microbiome as a Regulator of the Epidermal Barrier

In contrast to the intestinal epithelium, the epidermal barrier of the skin is markedly more complex, perhaps unsurprisingly given that it protects the body from both environmental challenges and pathogen invasion. The skin bears a large microbial community of its own, a complex collection of bacteria, archaea, fungi and viruses colonising all levels of the skin, from its outermost surface down to the dermis itself [98], though the dermal microbiota notably shows considerably more inter-individual compositional preservation than that of the epidermis [99]. The microbiota has been linked to numerous aspects of skin biology, and is required for skin health [100], both by competitively preventing pathogen colonisation [101] and, as we are increasingly realising, by directly influencing epidermal barrier function.

### 3.1. Epidermal Barrier Structure

In contrast to the simple columnar epithelium seen in the gastro-intestinal tract, the skin has a complex, multi-component structure. It can be divided into two main layers: the lower dermis, consisting of fibroblasts, mast cells and macrophages that are embedded within a complex proteinaceous extracellular matrix, which provides tensile strength and cushioning; and the external epidermis, an avascular, stratified squamous epithelium composed of keratinocytes embedded within an extracellular lipid matrix that serves as a barrier against invasion of exogenous chemical and biological agents and the loss of endogenous molecules.

The epidermis is notable for its multi-laminate structure, formed by accumulated layers of externally migrating epithelial cells at progressive degrees of differentiation (Figure 2). The innermost layer, the stratum basale, is formed from a single layer of proliferative columnar basal cells, connected to each other and the overlaying stratum spinosum through desmosomes and hemidesmosomes (structures superficially similar to desmosomes but which form direct connections to the extracellular matrix rather than to other cells [102]). Basal cells divide to form the polyhedral, desmosome-interconnected keratinocytes of the stratum spinosum. As cells migrate outwards through this layer towards the more exterior stratum granulosum, they begin to express a complex and changing pattern of keratin genes, with resultant keratin fibrils becoming increasingly interconnected through the actions of cross-linking proteins such as filaggrin, loricin and involucrin found in the dense keratohyalin granules that name the layer [103]. Above the stratum granulosum is the outermost layer of the epidermis, the stratum corneum, which is composed of up to 100 layers of highly keratinised, enucleated corneocytes embedded in a complex lipid matrix and provides resistance to mechanical shear, molecular transit and microbial invasion [104]. Migration of differentiating cells from the stratum basale basal to the surface of the stratum corneum takes approximately two weeks, whereupon individual corneocytes are lost imperceptibly to the environment through an enzymatically governed process termed desquamation [105,106]. Notably, as cells transit from the stratum granulosum to the stratum corneum they secrete large amounts of ceramides, cholesterol, fatty acids, and cholesterol esters into the extracellular matrix, adding to the hydrophobic barrier properties of the tissue [107].

For a long time, these secreted lipids were thought to be the major drivers of skin barrier function, despite the discovery, over 50 years ago, that neighbouring keratinocytes exhibited apparent membrane fusion just prior to cornification, at a point corresponding to the innermost diffusion limit of applied tracers [108], both signs suggesting the presence of junctional complexes. More recently however, the importance of TJs has been recognised, originating primarily from the discovery that claudin-1 null mice died shortly after birth due to excessive cutaneous dehydration caused by the lack of efficient skin barrier function [109]. Subsequent analysis of TJ molecule expression has identified the presence of numerous components, including claudins-1, -4, -6, -7, -11, -12 and -18, occludin, ZO-1, ZO-2 and cingulin, alongside ultrastructural evidence of TJ formation within the stratum granulosum [110,111,112,113], matching the position of the functional diffusion barrier [114]. Intriguingly, the formation of TJs appears to be very tightly restricted within the stratum granulosum, with only two layers of cells expressing full TJs [113] and only fragmentary TJs being found below this point. In contrast, once cells migrate into the stratum corneum and become fully keratinised, junctional components become fixed and no longer recyclable, becoming mere echoes of previous dynamics [115].

Besides TJs, the other major cell–cell junctional complexes of the epidermis have also been described. AJs have been found both at the dermis–epidermis junction and throughout the epidermal layers [116,117], although expression seems to favour the upper layers of the epidermis over the stratum basale [117]. The protein constituents of AJs and their intra/intercellular distributions are very similar to those seen in the intestinal epithelium, but they appear to be of even greater importance in enabling TJ formation in this tissue than they are in the gut. Specific deletion of E-cadherin within the murine epidermis has resulted in perinatal death due to dehydration and the inability to form an effective water barrier [118], highly reminiscent of the phenotype of claudin-1 null animals [109]. This phenotype was associated less with a loss of TJ molecule expression but rather with a failure in their organisation, particularly notable in terms of disrupted occludin, claudin-1 and ZO-1 architecture within the epidermis; significantly, both desmosomes and cornification appeared to be preserved.

Desmosomes are found throughout the viable layers of the skin, although their composition does appear to vary with location within the epidermis, i.e., desmoglein-3 and desmocollin-3 predominate in the strata basale and spinosum whereas desmoglein-1 and desmocollin-1 are found in more superficial layers [119]. As with their counterparts in the intestinal epithelium, epidermal desmosomes serve to connect cytoskeletal intermediate filaments with the plasma membrane and then to link adjacent cells. There are notable differences between intestinal and epidermal desmosomes however, in that epidermal structures express a much wider range of desmoglein, desmocollin and plakophilin family proteins, in differentiation stage-dependent patterns [120]. Why this is the case is unclear, but it is notable that desmosome strength also varies across the epidermis, with suprabasal desmosomes having significantly greater intercellular bonds than those of the stratum basale [121]. Another key difference in epidermal desmosomes occurs as maturing keratinocytes move into the stratum corneum, whereupon the layered intercellular structure of the desmosome is lost and the intracellular plaque becomes cross-linked to and incorporated within the cornifying membrane [122,123]. At this point, the protein corneodesmin becomes expressed alongside desmoglein-1 and desmocollin-1 in the extracellular part of the desmosome, forming homophilic connections with corneodesmin on adjacent cells, and marking the desmosome as a new structure, the corneodesmosome [124].

The different cell–cell junctions found within the epidermis act together with the lipid-rich extracellular matrix to form an effective barrier between the body and the external environment, most powerfully shown in the numerous skin diseases in which barrier failure is a significant component [125]. In rather marked contrast to the gut epithelium, investigation into whether the abundant skin microbiota can influence these structures is still at an early stage, but there is nonetheless evidence that this might be the case.

### 3.2. Microbial Influences on Skin Integrity

While there is significant evidence showing skin microbes to influence the function of the local immune response and hence cutaneous health (reviewed in detail in [126]), and there is evidence linking microbial functions with the lipid components of the epidermal barrier [127], direct evidence of whether microbes can modify barrier function through influences on cell–cell junctions components is only beginning to be revealed. As with the gut, evidence in principle of a role for skin microbes in regulating cell–cell junctions comes from analysis of germ-free animals. Skin from gnotobiotic mice was characterised by relatively subtly histological changes in the stratum corneum, accompanied by a more notable down-regulation in expression of the TJ markers claudin-1 and ZO-3 and the desmosome component desmoglein-1, changes underpinning deficits in barrier function [128]. Moreover, treatment with either a topical agonist to the aryl hydrocarbon receptor (AhR) or human commensal microbes could reverse defective barrier integrity [128]. Similarly, probiotic bacteria have been shown to ameliorate pathogen-induced barrier disruption, again acting via the AhR to induce re-expression of both keratinising factors such as filaggrin and loricin and the TJ component claudin-1 [129]. Whilst the activating factor for the AhR was not established in either of these studies, it is notable that a wide range of indoles, produced by microbial degradation of the amino acid tryptophan, are agonists at this receptor [130,131].

Skin-associated microbes are capable of metabolising surface lipids to produce agents able to modify barrier integrity. For example, the commensal *Staphylococcus epidermidis* has been shown to convert glycerol into lactic acid, a molecule which when applied to in vitro 3D primary skin cultures increased expression of filaggrin and sphingomyelin phosphodiesterase-1 (SMPD-1), both of which play important roles in regulating the skin barrier [132]. Filaggrin is required for keratinisation, promoting keratin fibre cross-linking while SMPD-1 is required for production of ceramide, an important constituent of the lipid extracellular matrix in the stratum corneum. Interestingly, succinic and lactic acids produced by *S. epidermidis* can inhibit the growth of the potentially pathogenic *Staphylococcus aureus* [132,133], an effect replicated by direct administration of the SCFA propionate [134], a reminder that whilst microbial metabolites may influence host features, the host response to such molecules may well be orthogonal to their role in bacterial physiology and ecology.

Direct studies of skin microbe-derived SCFAs as potential regulators of TJ function, as has been observed in the gut epithelium, are scarce, although it has been shown that SCFA administration could influence histone acetylation in primary human keratinocytes [135], reminiscent of effects seen on the intestinal epithelium [80,86], leading to significant changes in cytokine and immune mediator expression, although changes in junctional components were not reported. Further supporting a potential role for local microbe-derived SCFAs, the epidermis can respond to SCFAs produced by gut resident bacteria, with mice either on a high-fibre diet or directly fed butyrate showing significantly lower sensitivity to skin allergen exposure [136]. This effect appeared to be due to increased stratum corneum thickness coupled with improved barrier function, attributed to augmented expression of the keratin filament cross-linking protein loricin and greater cholesterol and ceramide production, but again direct analysis of junctional proteins was not performed.

Although skin microbes clearly influence several aspects of epidermal physiology, most notably the immune response and lipid contribution to barrier function, there remains considerable scope for investigation into their interactions with the different cell–cell junctional complexes found in this tissue. While the reason for this is uncertain, perhaps related to the historic focus on lipids as the primary mediator of skin barrier function, analysis of how TJs, AJs and desmosomes can respond to microbial actors may be of great importance in furthering both our understanding of cutaneous disease and its potential treatment.

## 4. The Blood–Brain Barrier and the Gut–Brain Axis

In contrast to the intestinal epithelium and the skin which bear their own microbiota and are thus exposed directly to microbial actions, the blood–brain barrier (BBB) represents a sterile physiological interface. Despite its physical separation from microbes however, there is increasing evidence that the BBB is a significant target for microbial actions, representing an important interface in microbe–brain communication, sometimes termed the gut microbiota–brain axis.

### 4.1. Structural Elements of the BBB

As the primary interface between the brain and the circulation, the BBB acts as a gatekeeper for blood-borne cells and molecules, protecting the delicate micro-environment of the brain tissue from their undue influence, as occurs in numerous metabolic and inflammatory diseases, including such major conditions as stroke, Alzheimer’s disease and multiple sclerosis [137,138]. This vital function is ultimately dependent upon the complex structure of the BBB, formed as it is from several distinct but integrated elements. The primary face of the BBB, the cerebromicrovascular endothelium, is similar in many ways to other endothelia within the body but possesses several characteristic features (Figure 3). First is the absence of fenestrations within the endothelial cells; communication can only occur through the cellular cytoplasm directly or via second messenger, or by paracellular routes. However, this route is itself limited by the second major feature of BBB endothelial cells, the presence of an extensive junctional complex composed of TJs and AJs, though not desmosomes [139], that essentially prevent uncontrolled cellular and molecular ingress into the brain [140,141]. Uptake of necessary nutrients into the brain is rather actively governed by the wide array of highly efficient influx and efflux transporters found within the endothelium, together acting to permit selective nutrient uptake and to actively remove metabolic waste products [142].

The endothelium, important as it is as the primary site of expression of TJ complexes within the BBB, is supported by numerous other cellular and non-cellular elements. Immediately adjacent to the endothelial cells is a complex basement lamina formed of four major glycoprotein family members, laminins, collagen IV isoforms, heparin sulphate proteoglycans and nidogens [143]. This basement lamina is actually composed of a pair of adjacent protein layers, produced by the endothelial cells and by perivascular astrocytes respectively, that whilst separate in larger vessels cannot be structurally distinguished at the level of the capillaries. The laminae can be discriminated by laminin complement however, with the endothelial basement layer containing laminin-411 and -511, whilst that derived from astrocytes is composed of laminin-111 and -211 [144]. This basement lamina is a functional as well as structural component of the BBB, being actively involved in communication and transport from the circulation to the neural parenchyma [145], and in the maintenance of TJ-mediated barrier integrity [146,147].

Two major cell types are found within the basement lamina, perivascular macrophages and pericytes, both of which play important albeit quite different roles in governing the BBB and its behaviour. Perivascular macrophages are the primary agents of immunosurveillance within the cerebral vasculature [148], but also facilitate glymphatic and intramural fluid drainage from the brain parenchyma to the circulation [149,150]. Interestingly there is also evidence that they may partially replace barrier function in the brain regions such as the area postrema that lack inter-endothelial TJs [151]. Pericytes in turn play a number of important roles within the BBB, including governing capillary diameter and hence cerebral blood flow distribution [152,153], regulating angiogenesis within the brain [154] and directly contributing to BBB integrity through modification of TJs [155,156].

The final major component of the BBB are the perivascular astrocytes, found on the parenchymal side of the basement lamina, which respond to pericytes-derived cues by fully enveloping blood vessels with extended processes, the so-called astrocyte end-feet [157]. These processes provide dynamic structural support to the BBB, both through production of the laminins that form a key part of the basement lamina [144], and through active promotion of inter-endothelial cell TJ formation [158,159]. Beyond this structural support, astrocytes functionally contribute to the regulation of substrate transport from the blood to the brain parenchyma and vice versa, actively taking up water through the channel aquaporin-4 [160], nutrients through a broad complement of nutrient transporters [159] and removing neuronal metabolic waste from the brain tissue to the blood for renal or hepatic clearance [161,162].

Together, these diverse structural and functional elements of the cerebral vasculature form the BBB and endow it with an extraordinarily strong barrier function. Free diffusion of all but the smallest molecules between the vasculature and the brain parenchyma is essentially prevented, allowing for tight homeostatic control of the brain’s micro-environment, and incidentally offering the opportunity to experimentally study BBB permeability through administration of different molecular weight tracers [163]. Despite this strength, the BBB is not a static structure but is rather highly plastic in response to challenge and demand with microbe-derived influences, among others, being powerful modulators of its function.

### 4.2. Regulation of BBB Integrity by Microbial Metabolites

Similarly to the gut epithelium, there are two major pathways by which microbial elements can affect the BBB in the absence of overt disease, either through the actions of microbial structural components or through those of microbe-derived metabolite. Of these, the effects of microbial components has received the greatest attention, with a substantial body of support having built up indicating that these agents can directly and indirectly regulate BBB integrity and thereby profoundly affect communication between the circulation and the brain.

That the BBB can be so targeted has been reported since the late 1950’s, with studies showing injection of rabbits with LPS to rapidly but temporarily increase access to the brain for co-administered tracers [164], with LPS treatment since becoming one of the most widely-used experimental models of BBB damage, despite its pleiotropic effects on the body and thus difficulty in interpreting exactly how it works. LPS has been shown to affect BBB integrity in several ways, including by modulating absorptive transcytosis [165], promoting immune cell adhesion and trafficking [166], and modifying expression of major efflux transporter systems such as P-glycoprotein [167,168]. Beyond these functional changes to the BBB, LPS can also directly disrupt cell–cell junctional complexes in the cerebrovascular endothelium, reducing expression of TJ components and JAMs [169,170,171]. The exact mechanism(s) underlying these effects of LPS remain uncertain, with evidence indicating roles for the CD14–TLR4 complex itself [172], MAP kinase-driven activation of matrix metalloproteases [173], stimulation of NADPH oxidase and production of reactive oxygen species [174], and indirect effects caused via systemic cytokine production [175]. Notably however, the increase in BBB permeability induced by acute LPS treatment is relatively short-lived [176], prompting the interpretation that changes in BBB function may be part of the adaptive response to inflammation/infection, and may be a trigger for physiological sickness behaviour and fever [177].

As with the intestinal epithelium and the epidermis, the first evidence that the BBB is a target for the actions of microbe-derived metabolites came from analysis of germ-free mice [178]. Development and maturation of the BBB was markedly compromised in these animals, with enhanced permeability to protein tracers apparent in both embryos and adults. Whilst vascular density and pericyte coverage was unaltered, germ-free mice showed significant TJ disruption, with reduced expression and altered localisation of both claudin-5 and occludin, though not ZO-1 in all brain regions examined. Supporting these findings, similar disruption in hippocampal expression of claudin-5 and occludin was seen in mice fed with non-adsorbed, broad-spectrum antibiotics [179,180]. Importantly, BBB disruption was ameliorated upon either colonisation of germ-free mice with a conventional murine microbiota, with either of two SCFA-producing bacterial strains, or upon feeding with a sodium butyrate solution [178], strongly implicating SCFAs as the principal mediating factor, akin to their actions upon intestinal epithelial TJs.

The idea that butyrate is beneficial is further supported by work showing that administration of high concentrations of the SCFA to protect against BBB damage in vivo limits both Evans blue tracer extravasation into the parenchyma and brain oedema in rodent models of traumatic brain injury [181] and ischaemic injury [182], in both cases providing notable protection when administered post-injury. Moreover, while analysis was not made of TJ molecules in the ischaemia study, the brain capillaries of mice that had received traumatic brain injury expressed markedly lower levels of occludin and ZO-1, changes which were significantly ameliorated by post-injury butyrate treatment [181].

The protective effects of butyrate in these studies were largely attributed to its role as an HDAC inhibitor, but there is evidence that SCFAs may protect at lower concentrations through their signalling at the G protein-coupled receptors FFAR2 and FFAR3. We have shown that the agonists at these receptors, butyrate and propionate, but not the low affinity SCFA acetate, could protect the barrier function of human cerebromicrovascular endothelial cells against in vitro inflammatory challenge [172]. In our hands, administration of physiologically relevant SCFA concentrations prevented disruption to TJ structure and hence barrier permeability through down-regulation of the LPS co-receptor CD14 and activation of the antioxidant master regulatory transcription factor Nrf2, again reinforcing the idea of a protective role for SCFAs. These effects of SCFAs have since been extended in the identification of downstream regulation by butyrate/propionate of cytoskeletal components and TJ localisation [183].

While most studies have focussed on the role of SCFAs, they are not the only class of microbe-derived molecule that are active at the BBB, with evidence suggesting that bile acids, methylamines and p-cresol conjugates are capable of influencing barrier permeability in vitro and in vivo. Bile acids are critically required for dietary lipid solubilisation and uptake [184], and are classed as either primary, produced by hepatic cholesterol metabolism, or secondary, where primary acids have undergone further metabolism by enteric microbes. Members of both classes of bile acid have been shown to damage BBB function, at both very high [185], and more physiologically relevant concentrations [186]. The primary chenodeoxycholic acid and the secondary deoxycholic acid both increased the permeability of the rat BBB in vivo and disrupted the expression pattern of occludin, ZO-1 and ZO-2 in rat brain microvascular endothelial cells in vitro [186]. Interestingly, this disruption was not due to changes in expression of protein or mRNA expression for these molecules but was rather driven by enhanced phosphorylation of occludin. In contrast, human brain microvascular endothelial cells treated in vitro with the secondary bile acid ursodeoxycholic acid were protected against bilirubin-induced permeability damage [187], suggesting that bile acid treatment is not purely negative. Further studies into the role(s) played by bile acids in governing BBB integrity are clearly warranted.

Several microbe-derived metabolites have been found to affect cardiovascular function, most prominently the dietary methylamines, trimethylamine (TMA) and trimethylamine N-oxide (TMAO). Levels of TMAO, derived by microbial processing of choline and L-carnitine to TMA and its subsequent oxidation in the liver, have been correlated with cardiovascular disease in numerous population-level studies (reviewed in [188]). Importantly though, not all population studies have replicated these links [189,190] and TMAO is protective in animal models of atherosclerosis [191], hypertension [192], non-alcoholic steatohepatitis [193] and impaired glucose tolerance [194]. In light of these discrepancies, we compared the effects of TMA and TMAO upon the BBB. Notably, we found marked differences between the effects of TMA and TMAO upon an in vitro model of the BBB endothelium, with TMA significantly enhancing endothelial permeability via disruption of both the actin cytoskeleton and ZO-1 distribution, indicative of damage to TJ complexes [195]. In contrast, TMAO enhanced both the cortical distribution of actin and ZO-1, acting through the mobilisation of annexin A1, a key TJ regulatory protein [196], leading to a greater permeability barrier. These effects of TMAO have been replicated in vivo, where pre-treatment of mice with methylamine protected BBB integrity in the face of both acute and chronic inflammatory challenge, effectively preserving cognitive function [195]. That the relatively beneficial TMAO is a host metabolic derivative of microbe-produced and considerably more detrimental TMA highlights the role of host processes in detoxifying potentially damaging metabolites.

Further evidence for the modulatory influence of host enzymes upon microbial metabolite effect comes from the study of *p*-cresol conjugates. Primarily, *p*-cresol is produced by microbial degradation of the aromatic amino acids tyrosine and phenylalanine within the gut, whereupon it crosses the intestinal wall into the portal vasculature [197]. Very little native *p*-cresol is found in the systemic circulation, rather it is rapidly and almost completely conjugated by host hepatic and enteric enzymes into *p*-cresol sulfate (pCS) and *p*-cresol glucuronide (pCG) [198] at a ratio of approximately 9:1 in humans or 1:1 in mice [199]. Interestingly, although both these conjugates can affect the BBB, their effects in vivo are essentially opposite in nature. Our studies of pCS identified potent permeabilising effects of this metabolite upon the BBB, acting through stimulation of the EGF receptor to trigger mobilisation of matrix metalloproteinases-2 and -9, damaging BBB integrity and inducing vascular leakage of macromolecules into the brain parenchyma [200]. In contrast, pCG had limited direct effects upon the BBB, but was able to almost completely prevent the permeabilising effects of either exogenous or circulating LPS in vitro and in vivo, acting through antagonism at the TLR4 receptor complex [201]. It seems highly likely that other such interactions exist between different host processing enzymes, microbial metabolites and/or microbial structural components, both at the BBB and other physiological barrier systems, indicating a vast scope for investigation and discovery.

## 5. Conclusions

Given the substantial differences in environment, function and cellular structure between the barrier systems of the body, it is striking how comparable their responses to microbe-derived molecule exposure are. In particular, it appears that development of physiological barriers is fundamentally dysfunctional in the absence of a microbiota, suggesting that the presence of microbial elements is an expected and normal part of physiology, and, moreover, that such microbe-derived molecules may have played an important role in the drive to generate these barriers in the first place. This further adds to the radical restructuring in our understanding of host–microbe dynamics that has occurred over the last several decades and emphasises the importance of considering both host and commensal influences on physiological systems and their behaviours in health and disease.

It is clear that a wide range of microbe-derived molecules can influence barrier function, and that the examples studied so far are likely to represent only the “tip of the iceberg”. Extensive research has been undertaken in order to understand the contribution of SCFAs to intestinal, cutaneous and blood–brain barriers for example, but even here most work has focussed on the three primary examples, acetate, propionate and butyrate. The contribution(s) made by other SCFA molecules and perhaps more importantly, their interactions with other microbial products have only just begun to be addressed.

Similarly, the question of which microbes produce the different metabolites shown to affect cell–cell junctions controlling physiological barriers remains largely unanswered. Certainly there are examples of individual bacteria known to produce specific metabolites (e.g., *Akkermansia muciniphila* and propionate [202], *Coriobacteriaceae* or *Clostridium* species and p-cresol [197]) but in general the identity of the microbes or groups of microbes able to produce most biologically active metabolites remains uncertain. This will be an important area to address in future research, particularly as the use of strategies such as probiotic treatment or faecal microbial transplantation moves into the clinic.

Of the different cellular structures responsible for cell–cell junctions and hence barrier strength, it appears that tight junctions in particular are most responsive to microbe-derived influences. While this may be due to their position as primary governors of junctional integrity, and thus the sites most likely to respond to microbial influences, it should be borne in mind that this may also be an artefact of what has been studied. Clearly, much work remains to be undertaken until we can truly say that microbial influences upon physiological barrier function are understood, and comprehensive examination of the different cell–cell structures will be an important target for future research.

In conclusion, given the ubiquity of microbes in the external world and the importance of maintaining a sterile internal environment to homeostasis, it is perhaps unsurprising that physiological barriers have evolved to respond to microbial influences. While this is most apparent in the intestinal epithelium, colonised as it is by a vast and diverse microbiota, accumulating evidence suggests this may be a more general feature of the other body barriers, whether, as with the skin, they bear their own microbiota, or similar to the BBB, they are typically sterile. As such, future studies of both microbe–epithelium interactions in other tissues, e.g., the lung, bladder, vagina and cervix, or of the remote effects of microbes upon sterile blood–tissue barriers may be highly relevant in understanding the physiology of these systems. Given the importance of physiological barriers in the range of human diseases, there is clearly a marked need to study the mechanistic links between microbes, microbe-derived factors and metabolites, and the regulation of cell–cell junctional complexes. By developing our understanding of these interactions, we will not only gain greater insight into the underlying biology of barrier function but have the potential to identify new therapeutic approaches and treatments for the many diseases characterised by barrier dysfunction.

## Figures and Tables

**Figure 1 life-13-00396-f001:**
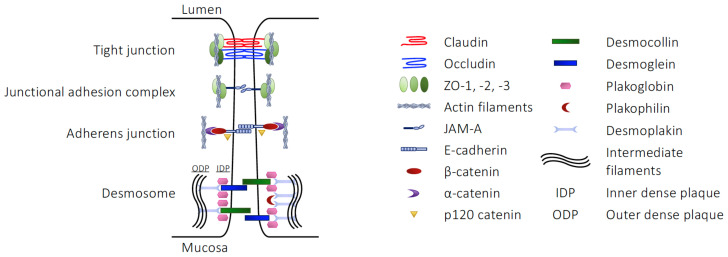
Cell–cell junctional molecules of the intestinal epithelium.

**Figure 2 life-13-00396-f002:**
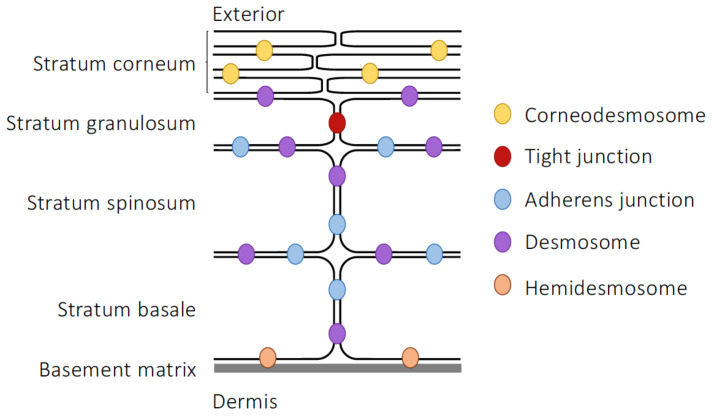
Structure and intercellular junctions of the epidermis.

**Figure 3 life-13-00396-f003:**
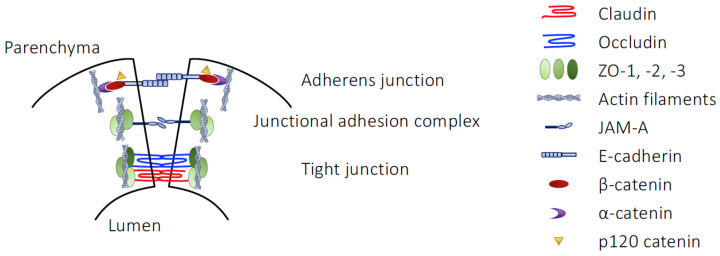
Junctional complexes of the cerebrovascular endothelium.

**Table 1 life-13-00396-t001:** Overview of the primary functions of cell–cell junction proteins.

Component	Junction Type	Junctional Role
Claudins	TJ	Homotypic/heterotypic cell–cell interactions
Occludin	TJ	Homotypic cell–cell interactions
Tricellulin	TJ	Homotypic cell–cell interactions at tripartite or greater junctions
ZO-1, -2, -3	TJ	Link between claudins, occludin or JAM-A and the actin cytoskeleton
JAM-A	JAM	Cell–cell communication, stabilisation of the TJ environment
E-cadherin	AJ	Homotypic cell–cell interactions
β-catenin	AJ	Act as a multipartite complex linking E-cadherin with the actin cytoskeleton
α-catenin	AJ	
p120-catenin	AJ	
Desmocollin	Desmosome	Cadherin family proteins, bringing neighbouring cell membranes into apposition
Desmoglein	Desmosome	
Plakoglobin	Desmosome	Catenin family proteins, linking desmocollin and desmoglein to desmoplakin
Plakophilin	Desmosome	
Desmoplakin	Desmosome	Link other desmosomal proteins to keratin intermediate fibres of the cytoskeleton

## Data Availability

Not applicable.

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
