# Peer review of "Regulation of Physiological Barrier Function by the Commensal Microbiota"

_life, 2023, doi:10.3390/life13020396_

Round 1

Reviewer 1 Report

Simon Mc Arthur provides an overview on the current knowledge on the microbiota as a barrier regulating factor. This is a timely and important topic. The manuscript provides an introduction to several relevant barrier mechanisms with focus on the gut, the skin, and the blood brain barrier. However, there are additional barriers that were not dealt with, e.g., the uterus, the lung, the bladder. The illustrations are clear and explanatory. Key references were thoughtfully chosen but the writing of the manuscript could be improved. Nevertheless, this is a highly relevant topic that is worth to be covered by a review article. I have some comments that should be addressed in form of a substantially revised manuscript:

1.      The introduction of this review is relatively short. More efforts should be made to explain why microbiota-dependent regulation of barrier function is a highly relevant question considering nutrition, immunity, circadian rhythmicity, development, and age of the host.

2.      To be comprehensive, also the uterus, the lung, the oral cavity and pharynx, the stomach, and the bladder should be dealt with. Otherwise, it should be specified in the abstract why the focus of this review was on the gut, the skin and the blood-brain barrier.

3.      On page 3 the distribution of claudin family members is discussed. Here, this review might benefit from a more detailed view on this topic.

4.      A table on antibiotics that directly affect elements of the gut barrier would be interesting. Are there effects on the gut barrier caused by antibiotics that are independent of their microbiota depleting effect?

5.      With regard to the implication of TLRs in gut barrier regulation, an overview on the studied conditions appears to be essential. Which studies focused on unchallenged conditions, which on diet conditions and metabolism, which on induced conditions of intestinal inflammation, which on infection etc. This might have a strong impact on the outcome and the conclusions that can be drawn. Therefore, a structured description along with a table will add. The same applies to the interesting topic of barrier-regulating microbial metabolites that were not described in much detail except for SCFAs. In general, the role of pathological barrier leakage should be separated from barrier regulation at healthy conditions.

6.      The regulation of gut barrier function by microbial and endogenous proteases and epithelial protease-activated receptors is a relevant topic that should be mentioned (see PMIDs: 34791291, 23115353, 31495063). This is a relevant topic for this review.

7.      Apart from epithelial barrier function, the article puts focus on the blood-brain barrier as a relevant endothelial barrier. There are numerous other relevant endothelial barriers, e.g. the gut-vascular barrier that has strong influence on selective nutrient uptake and infection control (see PMIDs: 26564856, 31419514, 30093598). This aspect can be included in the description of the intestinal barrier. With regard to vascular barrier function, a more detailed description of additional physiological methods to detect vascular leakage other than Evans blue extravasation would add. Moreover, a more structured explanation of the pathologies that are associated with a disruption of the blood-brain barrier is needed.

Author Response

I would like to thank the reviewer for their consideration and helpful comments. A detailed reply to their points is provided below:

Reviewer 1

Simon McArthur provides an overview on the current knowledge on the microbiota as a barrier regulating factor. This is a timely and important topic. The manuscript provides an introduction to several relevant barrier mechanisms with focus on the gut, the skin, and the blood brain barrier. However, there are additional barriers that were not dealt with, e.g., the uterus, the lung, the bladder. The illustrations are clear and explanatory. Key references were thoughtfully chosen but the writing of the manuscript could be improved. Nevertheless, this is a highly relevant topic that is worth to be covered by a review article. I have some comments that should be addressed in form of a substantially revised manuscript:

  1. The introduction of this review is relatively short. More efforts should be made to explain why microbiota-dependent regulation of barrier function is a highly relevant question considering nutrition, immunity, circadian rhythmicity, development, and age of the host.

The introduction to the review has been lengthened as suggested, providing greater clarity on why consideration of the role of the microbiota upon barrier functions is relevant, see lines 26 to 47.

  1. To be comprehensive, also the uterus, the lung, the oral cavity and pharynx, the stomach, and the bladder should be dealt with. Otherwise, it should be specified in the abstract why the focus of this review was on the gut, the skin and the blood-brain barrier.

Extensive consideration of all body barrier function will become cumbersome for the reader, given that this area of biology is still in its infancy. Rather, by considering the gut epithelium, the skin epidermis and the blood-brain barrier, I provide examples of barriers characterized by a high abundancy of microbes, a low microbial biomass and a microbe-free environment respectively, allowing greater comparison of these three micro-environments.  I have put this reasoning into the introduction (word space being limited in the abstract) to make this approach explicit (see lines 35 to 47), and have altered the conclusion to discuss more explicitly the potential lessons these studied examples might have for our understanding of other barrier systems, see lines 804-7.

  1. On page 3 the distribution of claudin family members is discussed. Here, this review might benefit from a more detailed view on this topic.

Greater detail is provided on this topic as suggested, see lines 123 to 135. 

  1. A table on antibiotics that directly affect elements of the gut barrier would be interesting. Are there effects on the gut barrier caused by antibiotics that are independent of their microbiota depleting effect?

The aim of this review is to directly address the ways in which microbes themselves can positively and negatively reflect physiological barriers, primarily through their secreted metabolites or structural components. Whilst discussion of direct pharmacological effects of antibiotics independent of their actions upon gut microbes is interesting, it would move into a rather different field and thus lies outside the review’s scope. Moreover, given that direct effects of antibiotics on gut epithelial cells is a rather poorly studied area, any coverage of this topic would be necessity be very speculative, hence I would rather not go down this route. I have however, expanded discussion of the impact of microbial depletion by antibiotic use and the information this provides regarding microbe-gut epithelial cell-cell junctions communication, see lines 225-239

  1. With regard to the implication of TLRs in gut barrier regulation, an overview on the studied conditions appears to be essential. Which studies focused on unchallenged conditions, which on diet conditions and metabolism, which on induced conditions of intestinal inflammation, which on infection etc. This might have a strong impact on the outcome and the conclusions that can be drawn. Therefore, a structured description along with a table will add. The same applies to the interesting topic of barrier-regulating microbial metabolites that were not described in much detail except for SCFAs. In general, the role of pathological barrier leakage should be separated from barrier regulation at healthy conditions.

The aim of this review was to specifically focus on the physiological actions of microbes/microbe-derived metabolites upon different barriers, rather than discussing the vast literature regarding the diverse pathologies of the gut, skin and cerebral vasculature which may involve microbial dysbiosis to a greater or lesser extent. Predominantly, the sources cited have investigated the direct effect of different microbial components in the absence of other complicating pathological phenotypes, and the text has now been altered in both the introduction and variously throughout the review to make this more apparent.

  1. The regulation of gut barrier function by microbial and endogenous proteases and epithelial protease-activated receptors is a relevant topic that should be mentioned (see PMIDs: 34791291, 23115353, 31495063). This is a relevant topic for this review.

I thank the reviewer for this excellent suggestion, and have now included a section discussing the role(s) of microbial proteases in physiological gut barrier control, see lines 282-313

  1. Apart from epithelial barrier function, the article puts focus on the blood-brain barrier as a relevant endothelial barrier. There are numerous other relevant endothelial barriers, e.g. the gut-vascular barrier that has strong influence on selective nutrient uptake and infection control (see PMIDs: 26564856, 31419514, 30093598). This aspect can be included in the description of the intestinal barrier. With regard to vascular barrier function, a more detailed description of additional physiological methods to detect vascular leakage other than Evans blue extravasation would add. Moreover, a more structured explanation of the pathologies that are associated with a disruption of the blood-brain barrier is needed.

As stated above, the three barriers chosen were selected as representative of those associated with high and low microbial biomass and sterile barriers, focusing at the level of intercellular junctions in line with the focus of the Special Issue. Moreover, while I certainly agree that it is an interesting topic, given that the regulation of the gut vascular barrier has largely been studied under pathological conditions, the information available lies outside the scope of this review, which specifically aimed to examine physiological control of barrier function by microbial elements.

The techniques described in the review as being used to detect vascular leakage at the blood-brain barrier are those employed by the studies quoted. A fuller consideration of how blood-brain barrier integrity can be monitored has been published elsewhere (Saunders et al. 2015 Front. Neurosci.) and is now cited to avoid drawing focus of the review away from its primary field of interest.

As with the gut barrier, the aim of this review is to discuss the role of microbes and their soluble mediators in the regulation of physiological barrier function as opposed to the allostatic conditions of disease. As such, lengthy discussion of disease states involving the BBB lies outside the scope of the review; I have rather directed the reader to some of the excellent recent reviews of this area.

Reviewer 2 Report

he review, entitled “Regulation of physiological barrier function by the commensal microbiota” had described that through comparison of the impact commensal microbes had on cell-cell junctions in three key physiological barriers, the gut epithelium, the epidermis and the blood-brain barrier, this review emphasized the important contribution microbes and microbe-derived mediators play in governing barrier function. The author has provided a well-organized and detailed manuscript, but there are still some minor issues that need to be revised and my suggestion is minor revision.

Comments:

1. In the part of “2.2 Microbial influences on the intestinal epithelial barrier structures”, “3.2 Microbial influences on skin integrity”, “4.2 Regulation of BBB integrity by microbial metabolites”, it would be better and meaningful if the author provides some information about the genus or species that the microorganisms play the role in these processes. Please give some example strains in these sections.

2. Please try to use tables to show the data related to the pathway or proteins which are important in separating the external environment from the internal, such as TJs, JAMs, and AJs.

3. What will happen when the microbial community at the barrier is disturbed? Could the author supplement related information?

Author Response

I would like to thank the reviewer for their consideration and helpful comments. A detailed reply to their points is provided below:

The review, entitled “Regulation of physiological barrier function by the commensal microbiota” had described that through comparison of the impact commensal microbes had on cell-cell junctions in three key physiological barriers, the gut epithelium, the epidermis and the blood-brain barrier, this review emphasized the important contribution microbes and microbe-derived mediators play in governing barrier function. The author has provided a well-organized and detailed manuscript, but there are still some minor issues that need to be revised and my suggestion is minor revision.

Comments:

  1. In the part of “2.2 Microbial influences on the intestinal epithelial barrier structures”, “3.2 Microbial influences on skin integrity”, “4.2 Regulation of BBB integrity by microbial metabolites”, it would be better and meaningful if the author provides some information about the genus or species that the microorganisms play the role in these processes. Please give some example strains in these sections.

While I agree that this would certainly be useful information to have, unfortunately in the vast majority of cases which microbial species or group of species act to produce individual metabolites is largely unknown and the subject of substantial and active microbiological study. Thus, while I wish I could provide this information, this is not really possible without extensive tangential discussion lying beyond the scope of the current review. In lieu of this information, I have added to the conclusions of the review to emphasize the point, see lines 780 to 787.

  1. Please try to use tables to show the data related to the pathway or proteins which are important in separating the external environment from the internal, such as TJs, JAMs, and AJs.

The roles of the different protein components of TJs, JAMs, AJs and desmosomes are now summarized in Table 1 as suggested.

  1. What will happen when the microbial community at the barrier is disturbed? Could the author supplement related information?

To an extent, some of this information is already present within the review in discussion of the effects of germ-free development upon the different barrier systems. I have however extended this as suggested to consider the impact of antibiotic-mediated microbial disruption in the gut (see lines 225 to 239) and, albeit in a much more limited way due to a lack of studies, on the blood-brain barrier (see lines 670 to 671). Despite intensive searching, I cannot find any direct studies of how antibiotics can affect cell-cell junctions in the skin outside of the context of infection, so this area is necessarily restricted to consideration of the microbial disturbance caused by germ-free development.

Round 2

Reviewer 1 Report

The author addressed all my comments.